# The Primary Cilium of Adipose Progenitors Is Necessary for Their Differentiation into Cancer-Associated Fibroblasts that Promote Migration of Breast Cancer Cells In Vitro

**DOI:** 10.3390/cells9102251

**Published:** 2020-10-08

**Authors:** Pascal Peraldi, Annie Ladoux, Sophie Giorgetti-Peraldi, Christian Dani

**Affiliations:** 1Faculté de Médecine, Université Côte d’Azur, CNRS UMR7277, Inserm U1091, IBV, 06107 Nice, France; Annie.LADOUX@univ-cotedazur.fr (A.L.); Christian.DANI@univ-cotedazur.fr (C.D.); 2Team Cellular and Molecular Physiopathology of Obesity, Université Côte d’Azur, Inserm, C3M, Bâtiment Universitaire ARCHIMED, 06204 Nice, France; Sophie.GIORGETTI-PERALDI@univ-cotedazur.fr

**Keywords:** adipose progenitor, primary cilium, cancer associated fibroblasts, breast cancer cells

## Abstract

Cancer associated fibroblasts (CAFs) are central elements of the microenvironment that control tumor development. In breast cancer, CAFs can originate from adipose progenitors (APs). We, and others, have shown that the primary cilium, an antenna-shaped organelle, controls several aspects of APs’ biology. We studied the conversion of human APs into CAFs by breast cancer cell lines (BCCs). Deletion of the cilium of APs by a pharmacological inhibitor, or by siRNA, allow us to demonstrate that the cilium is necessary for the differentiation of APs into CAFs. BCCs increase production of TGF-β1 by APs, which is a known inducer of CAFs. Pharmacological inhibition of TGF-β1 signaling in APs prevents their conversion into CAFs. Since we previously showed that deletion of the APs’ cilium inhibits TGF-β1 signaling, we propose that BCCs induce TGF-β1 production in Aps, which binds to the primary cilium of Aps and leads to their differentiation into CAFs. Inhibition of APs conversion into CAFs induces a loss in some of the biological effects of CAFs since deletion of the cilium of APs decreases their effect on the migration of BCCs. This is the first observation of a function of the cilium of APs in their conversion into CAFs, and its consequences on BCCs.

## 1. Introduction

The tumor microenvironment is an essential element of cancer development. It is composed of macrophages and immune cells responsible for the production of various cytokines and growth factors [1]. Cancer cells induce the differentiation of some cells of the stroma into cancer associated fibroblasts (CAFs) [2,3,4,5]. These CAFs have various functions. At first, during the development of the tumor, CAFs can exert anti-tumor properties. However, later on, their effect become pro-tumorigenic. Several studies have linked the function and the number of CAFs to the outcome of the disease [3]. CAFs produce extracellular matrix and growth factors that modulate proliferation of cancer cells as well as metastasis and resistance to chemotherapy.

Because of their functions in tumor progression, CAFs are a focus of attention for the development of strategic therapies against cancer. However, CAFs are heterogeneous cells and their precise characteristics, as well as the identity of their precursors, are not clearly defined. Depending on the cancer studied, and, within the same tumor, several precursors of CAFs have been identified [2,3,5]. They have been shown to arise from epithelial-mesenchymal transition of cancer cells. They are derived from normal fibroblasts of the stroma or they originate from differentiation of mesenchymal stem cells. Furthermore, there is no specific markers for CAF. Instead, several populations of CAFs have been described, which expressed a different type of markers. These markers do not necessarily overlap between the different populations, and they can be found in other non-cancer related cell types. The CAFs populations differ according to the type of tumor and, within the same tumor, various types of subpopulations exist [6,7,8,9]. Some researchers are classifying CAFs according to their genotype and function. Thus, using techniques such as single cell RNA sequencing or FACS, authors are classifying CAFs as “myofibroblast (or myo) CAFs,” “inflammatory CAFs,” “vascular CAFs,” “cycling CAFs,” “developmental CAFs” [3,8]. So far, no unified classification has been adopted. Thus, cells are considered as “myo CAFs” if they are in proximity of a tumor, express α-smooth muscle actin (α-SMA), secrete extracellular matrix (ECM) proteins, and produce TGF-β1. In breast cancer, these “myo CAFs” are particularly abundant and produce the desmoplastic response: a dense collagenous stroma responsible for the clinical presentation of a tumor as a ‘lump.’ As indicated by their name, “myo CAFs” resemble myofibroblasts and cells that appear after tissue lesion and are characterized by α-Smooth Muscle Actin (α-SMA) expression and secretion of ECM proteins [10,11].

We have reported that the differentiation of adipose progenitors (APs), which are a type of mesenchymal stem cells, into myofibroblasts is dependent upon the primary cilium [12]. APs can also differentiate into CAFs and are found in several organs prone to cancer [13,14]. APs are particularly relevant in breast cancer since the adipose tissue is abundant in this organ and has been shown to be in direct contact with tumors. The primary cilium is an antenna-shaped organelle present in nearly all cells of the organism [15,16,17]. It is a post-mitotic organelle that grows from the mother centrioles of the centrosome. It is formed by an axoneme composed of nine doublets of tubulin. The cilium acts as a mechanoreceptor in kidney cells and is also involved in the transduction pathway of several molecules such as Hedgehog (Hh), Wnt, and TGF-β1. Thus, it plays an important role in embryogenesis and cell differentiation.

We have investigated if the cilium was involved in the conversion of APs into CAFs after incubation with breast cancer cell lines (BCCs). Using a simple semi co-incubation technique that allows to analyze independently two cell types, we observed that two well-characterized BCCs (MCF-7 and T47D cells) convert APs into CAFs by a mechanism dependent upon the primary cilium and TGF-β1. Deciliated APs, which are unable to differentiate into CAFs, lose their ability to stimulate migration of cancer cells but do not lose their effect on BCCs proliferation.

This is the first observation that differentiation of APs into CAFs is dependent upon the primary cilium and could have implication for the design of strategic therapies against the development of cancer.

## 2. Materials and Methods

### 2.1. Material

HPI-4, used at 60 µM for 24 h, was from Sigma-Aldrich (Saint Quentin Fallavier, France). TGF-β1 was from PeproTech France (Neuilly-Sur-Seine, France) and was used at 5 ng/mL. Antibodies: anti-α-SMA (A5228), a mouse monoclonal antibody raised against an NH2 terminal synthetic decapeptide of smooth muscle actin and anti-acetylated Tubulin (T6793) in which a mouse monoclonal was raised against an epitope located within four residues of Lys-40 when this amino acid is acetylated, and were from Sigma-Aldrich. Anti-Fibronectin (sc-8422), a mouse monoclonal antibody raised against a T cell leukemia biopsy of human origin, anti—Vimentin (sc-6260), a mouse monoclonal raised against purified Vimentin from porcine eye lens, and anti-ARL13B (sc-515784), a mouse monoclonal raised against amino acids 414–428 in the c-terminus of human ARL13B, were from Santa Cruz biotechnology (Heidelberg, Germany). Anti-Pericentrin (A301-348A), which is a rabbit polyclonal antibody raised against amino acid 175–225 of Pericentrin, was from Bethyl (Euromedex, Souffelweyersheim, France). Anti-E-Cadherin (3195), a rabbit polyclonal raised against a synthetic peptide corresponding to the sequence surrounding Pro780 of human E-cadherin protein, was from Cell Signaling Technology (Ozyme, Saint-Cyr-L’École, France). Secondary antibodies coupled to Alexa Fluor 488 or 647 were from Life Technologies (Saint Aubin, France). Secondary antibodies coupled to horseradish peroxidase were from Promega (Charbonnières-les-Bains, France).

### 2.2. Cell Culture

#### 2.2.1. APs

Establishment of human APs from adipose tissue have been previously described [18]. Breast adipose progenitors were prepared as described in Reference [19]. They were derived from the stroma vascular fraction of women who underwent breast reduction procedures. Patients received clear information before surgery and all of them gave written and signed consent. Tissue sections were obtained from res nullus from surgeries performed on healthy donors with the approval of the Centre Hospitalier Universitaire de Nice Ethical Review Board (L1235-2 and L1211-2), which is in accordance with the rules of the French Regulatory Health and with the Declaration of Helsinki. APs were grown in Dulbecco’s modified Eagle’s medium (DMEM) supplemented with 10% fetal calf serum (FCS), 5 ng/mL hFGF-2, 60 μg/mL penicillin, and 50 μg/mL streptomycin. At day 2 post-confluence (day 0), cells were maintained in “depletion medium,” i.e., DMEM without phenol red supplemented with 10 µg/mL transferrin, 60 μg/mL penicillin, and 50 μg/mL streptomycin. 

#### 2.2.2. BCCs

Michigan Cancer Foundation-7 (MCF-7) and T47D were grown in Dulbecco’s modified Eagle’s medium (DMEM) supplemented with 10% FCS, 60 μg/mL penicillin, and 50 μg/mL streptomycin.

### 2.3. siRNA

APs were seeded on cover slips. After 24 h, they were transfected with a siRNA directed against IFT88 or with a non-relevant siRNA using HiPerfect Transfection Reagent (Qiagen, Courtaboeuf, France) as recommended by the manufacturer. The transfection was replicated three days later, and experiments were performed after another three days. Sequences of siRNA against IFT88 were IFT#1: CCGAAGCACUUAACACUUA; IFT#2: CUGAAACUUCACGCAAUCC, as described in Reference [20].

### 2.4. Immunocytochemistry

APs were seeded on glass coverslips and treated as described in the text. Cells were rinsed with PBS and fixed with Roti-Histofix (Roth, Lauterbourg, France) for 20 min at room temperature. Fixed-cells were incubated in PBS with 3% goat serum, 0.1% tween-20, and 0.1% triton X-100 for 30 min at room temperature. Cells were incubated with the appropriate antibodies in the same buffer for 90 min at room temperature. After three washes in PBS, coverslips were incubated with the appropriate secondary antibody coupled to AlexaFluor (1:600) for 45 min at room temperature. Cells were mounted in mowiol. Images were taken on an Axio Observer microscope (Carl Zeiss, Le Pecq, France) with an EC Plan Neofluar 40X (NA 1.3) oil objective using AxioVision 4.8.2 software.

### 2.5. Wound Healing Assay 

APs were seeded on cover slips (18 mm in diameter) and BCCs on six well dishes. After 24 h in depletion medium, APs were treated or not with HPI-4 for 24 h. HPI-4 was removed, and cover slips were placed on BCCs. Controls, with cover slips without APs, were performed. After four days, culture media were recovered, cover slips were removed, and BCCs were trypsinised and replated on two 24-wells in growth medium, in the presence of mitomycin C (10 µg/mL). After 3 h, cells were placed in their original medium and scratched with a pipet tip. Pictures were taken just after the scratch and after 24 h. Localisation was performed by drawing lines at the bottom of the plate. Pictures were taken on the Axio Observer microscope (Carl Zeiss, Le Pecq, France) with an EC Planar neofluar 5X (NA 1.3). The width of the scratch was measured using FiJi [21]. For each condition, four wells were studied and two measures by wells were performed. A diagram of the protocol is provided in Appendix A.

### 2.6. Western Blot 

Cells were washed with ice-cold phosphate-buffered saline (6 mM Na_2_HPO_4_, 1 mM KH_2_PO_4_, pH 7.4, 140 mM NaCl, and 3 mM KCl) and lysed in RadioImmunoPrecipitation Assay (RIPA) buffer (50 mM Tris pH 7.5, 150 mM NaCl, 1% NP40, 0.1% SDS, 0.5% Na Deoxycholate, 5 mM NaF, 2.5 mM Na_4_P_2_O_7_, and Complete Protease Inhibitor Cocktail (Roche Diagnostics, Meylan, France) for 20 min at 4 °C. Lysates were centrifuged (8000 g) for 10 min of 10 μg of proteins that were resolved on SDS-PAGE under reducing conditions on a 10% polyacrylamide gel. Proteins were transferred to a polyvinylidene difluoride membrane (Merck Millipore, Fontenay sous Bois, France), and Western blot analysis were revealed using a Amersham Imager 600 v1.0 imaging system.

### 2.7. RNA Extraction and Analysis

Total RNAs were extracted with the TRI-Reagent kit (Euromedex, Souffelweyersheim, France), according to the manufacturer’s instructions. Total RNA was subjected to real-time quantitative reverse transcription (RT)-polymerase chain reaction (PCR) analysis, as described in Reference [22]. Primers were designed using Primer Express software (Applied Biosystems, Courtaboeuf, France) and validated by testing PCR efficiency using standard curves (85% ≤ efficiency ≤ 115%). Gene expression was quantified using the comparative C_T_ (threshold cycle) method on a StepOnePlus system (Applied Biosystems). TATA box Binding Protein (TBP) expression was used as a reference. A list of the primers used for the Real Time Quantitative Polymerase Chain Reaction (RT-QPCR) is provided in the Appendix A.

### 2.8. Statistics

Data are shown as means +/− SD. Statistically significant differences between groups were analyzed using Student’s t-test or ANOVA using graphPad InStat 3.02.

## 3. Results

### 3.1. A Semi Co-Incubation Technique to Study APs Conversion into CAFs

The conversion of APs into CAFs by BCCs is an important process in tumor progression. Since we previously showed that the primary cilium controls several aspects of APs biology [12,22,23], we studied the function of the cilium in the differentiation of APs into CAFs, induced by BCCs.

First, we verified that our well-characterized model of APs [18] could differentiate into CAFs. We co-cultured APs in the presence or in the absence of MCF-7, which is a classical study model for BCC. After four days, cells were analyzed by immunofluorescence using α-SMA (in green), which has been established as the more reliable marker for CAFs [2,3] and E-Cadherin (in pink). This is a specific epithelial cell marker that labelled BCCs (Figure 1A). In control conditions, α-SMA was expressed at a low level in APs. Co-incubation of APs with MCF-7 increased α-SMA labeling of APs, indicating that MCF-7 converted APs into CAFs. Labeling with ARL13b (in green), a marker of the primary cilium and Pericentrin (in red), which is a marker of the basal body, revealed that CAFs, in proximity to MCF-7, were ciliated (Figure 1B).

This co-culture result confirmed that incubation of APs with MCF-7 converted APs into CAFs. Previous studies used conditioned medium (CM) to reach the same conclusion. Although they are widely used and have provided important information, these methods have a few drawbacks. Classical co-incubation techniques do not allow for an easy quantification of gene and protein expressions, specifically, in each type of cell. Cells can be studied after trypsinization and cell sorting, but this requires additional steps that can affect gene and protein expression. The use of conditioned medium has a limitation. If short-lived molecules are involved in the process, a frequent resupply of conditioned medium is required. Moreover, if feedbacks loops take place between the two cells lines, they will not be revealed when using conditioned medium.

We decided to use a technique of a “semi” co-culture. Cancer cells were grown on the cell culture dishes and APs, separately, on cover slips. When cells reached confluence, cover slips with APs were placed on top of the cancer cells. This allows cells to be incubated constantly in the presence of the secretome of both cell types. Controls were performed using cover slips without AP. After 4 days, cover slips were removed and analyzed.

We compared the efficiency of this technique to incubation with conditioned medium and investigated for a possible contamination between cells types. We grew two classical breast cancer cell lines MCF-7 and T47D to confluence. Cover slips containing APs were incubated in depletion medium, co-incubated with BCCs, or incubated with conditioned medium from MCF-7 or T47D (changed every day) for four days. Proteins were extracted from APs and BCCs and analyzed by the Western blot (Figure 1C).

We studied α-SMA expression to evaluate the conversion of APs into CAFs. Incubation of APs with conditioned medium from cancer cells increased α-SMA expression. α-SMA induction was more potent when the experiment was performed using the “semi” co-incubation technique. Compared to other studies, the relative low increase in α-SMA expression after treatment of APs with conditioned medium is likely due to the fact that the treatment here was shorter (4 days vs. 30 days) [24].

To determine the level of cross contamination between APs and BCCs, we performed Western blots using specific markers for epithelial cell (E-Cadherin) or APs (Vimentin). As observed in Figure 1C, Vimentin is expressed only in APs while E-Cadherin is detected only in BCCs extracts indicating that the level of cross contamination is barely detectable.

We used Vimentin as the loading control for Western blots. It is an established marker of APs and it has previously been reported that its expression is not modified after co-incubation with BCCs [25]. Furthermore, compared to classical loading controls, such as ERK or Tubulin (Figure 1C), Vimentin is not expressed in BCCs. Thus, in the case of an accidental contamination of APs by BCCs during the preparation of cell extracts, Vimentin provides a better loading control than other proteins, which are also expressed in BCCs.

These data show that BCCs convert APs into CAFs and that the technique of “semi” co-incubation allows for a specific analysis of gene and protein expression in each cell type.

### 3.2. Conversion of APs into CAFs by BCCs Is Dependent upon the Primary Cilium

The function of the cilium in the differentiation of APs into CAFs was first determined by deleting the primary cilium using Hedgehog Pathway Inhibitor 4 (HPI-4). HPI-4 (also known as ciliobrevin A) is a specific inhibitor of dynein in vitro and a well-described inhibitor of the cilium formation [26,27]. APs were grown to confluence on the cover slip, incubated with depletion medium for 24 h and, then, treated with HPI-4 for another 24 h. Ciliation of APs was assessed by making a dual labeling of the cells with ARL13B and Pericentrin (Figure 2A). ARL13B is a marker of the axonem of the primary cilium and Pericentrin is a marker of the basal body, which serves as a nucleation site for the growth of the axonem microtubules. Percentage of ciliated cells are calculated by the number of cilia when compared to the number of basal bodies. As we previously reported [12], HPI-4 induces a decrease in ciliated cells. Labeling of primary cilia using acetylated Tubulin as a marker of the axonem gives similar results (Appendix A). After HPI-4 treatment, cover slips were incubated with BCCs for four days in a medium without HPI-4. Western blots were performed on protein extracts from APs using antibodies against α-SMA or Fibronectin (FN), which is another marker of CAFs (Figure 2B). Incubation of APs with BCCs increased the expression of α-SMA and FN. As observed through α-SMA and FN expression, the ability of BCCs to convert APs into CAFs was decreased when APs were deciliated. An immunofluorescence experiment (Figure 2C) using α-SMA and FN confirmed that deciliation of APs decreased their conversion into CAFs.

CAFs are a heterogeneous population of cells with different genes’ expression and functions. We studied the expression of other genes associated with CAFs, Col1A1, TGF-β1, PDGFR-α, FAP, Fibroblast-Specific Protein 1 (FSP1)/S100A4, and leptin by RT-QPCR (Figure 3). Col1A1, which is a gene associated with “myo CAF” [28], was increased after co-incubation with BCCs only if APs were ciliated. This profile is similar to the one observed for α-SMA and FN proteins expression. The expression of FAP and FSP1 was unchanged. PDGFR-αexpression was also not modified. Incubation of APs with BCCs increases TGF-β1 expression in APs and, as expected (see discussion), inhibition of the cilium in APs did not modify its expression. Leptin expression was increased in APs after incubation with BCCs. We did not observe a statistically significant difference if APs were ciliated or not. The expression of genes associated with preadipocyte such as Hoxc8 and Hoxc9 was not modified. As described in the discussion below, CAFs obtained in these experiments present the characteristics of a “myo CAF” population since they express α-SMA, FN, COL1A1, and TGF-β1.

The function of the cilium in CAFs conversion was verified using siRNA against IFT88. IFT88 expression is necessary for the primary cilium formation [29]. APs were treated with two siRNAs directed against IFT88 as described in “methods” and ciliation was analyzed through ARL13B and Pericentrin expression by immunofluorescence (Figure 4A). Under control conditions, 94% of APs were ciliated. Treatment with siRNA #1 decreased ciliation to 51% of the cells and siRNA #2 to 26% (Figure 4B). This was concomitant with a decrease in IFT88 mRNA expression (Figure 4C). APs treated with a control siRNA or siRNA directed against IFT88 were incubated with BCCs and analyzed by Western blot (Figure 4D) or immunofluorescence (Figure 4E) using α-SMA and FN antibodies, as described in Figure 2. Incubation of APs with BCCs increased APs conversion into CAFs. siRNA#1 and siRNA#2 inhibited this conversion even though siRNA#1 was less potent, especially concerning FN expression measured by the Western blot. The difference in efficiency between the two siRNAs was likely linked to their different potency to decrease APs ciliation.

These results indicated that the primary cilium of APs is necessary for their conversion into CAFs. We wanted to expand these results to other types of APs and in APs from breast adipose tissue (B-APs) that are particularly relevant for the study of CAFs in breast cancer. We seeded primary cultures from B-APs to confluence and we analyzed ciliation by labeling ARL13B and pericentrin (Figure 5A). As observed, most B-APs were ciliated and a 24-h treatment with HPI-4 decreased ciliation (Figure 5B). B-APs were grown on cover slips, treated or not with HPI-4, and incubated with BCC. After four days, expressions of α-SMA and FN were analyzed by Western blot (Figure 5C) and immunofluorescence (Figure 5D). Incubation of B-APs with BCCs induced their conversion into CAFs as observed through an increase in α-SMA and FN expressions. When B-APs were deciliated by HPI-4 before incubation with BCCs, expressions of α-SMA and FN sharply decreased. These data indicated that the primary cilium of B-APs was necessary for their conversion into CAFs. Because primary cultures of B-APs are not as resistant as the APs cells, we did not perform siRNA experiments with B-APs.

Together, these data present the first report underlying the function of the primary cilium in the differentiation of human APs from different origins into CAFs.

### 3.3. Conversion of APs into CAFs by BCCs Is Dependent upon TGF-β1

TGF-β1 is a known inducer of CAFs and is expressed at high levels in the plasma of breast cancer patients [2,3,4,30]. It has been established that the TGF-β1 receptor is located in the primary cilium [31] and we were previously shown that TGF-β1 signalization in APs was dependent upon the primary cilium [12].

First, we studied TGF-β1expression in APs and BCCs, alone, or after co-incubation. APs were grown on cover slips and incubated or not with MCF-7. After four days, RNA from APs or from MCF-7 were recovered, and TGF−β1expression was analyzed by RT-QPCR (Figure 6A). As was observed in Figure 3, co-incubation with BCCs increased the expression of TGF-β1 in APs. TGF-β1 expression was lower in MCF-7 than in APs, specifically after co-incubation. This suggests that most of the effect of TGF-β1 happened through an APs autocrine loop.

We then tested if TGF-β1 was involved in the differentiation of APs into CAFs. APs were grown on cover slips and incubated with MCF-7 or with TGF-β1 in the presence or in the absence of SB431542, which is an inhibitor of the TGF-β1 receptor. α-SMA and FN expressions were monitored by Western blots (Figure 6B) and immunofluorescence (Figure 6C) experiments. As expected, TGF-β1 induced an increase in α-SMA and FN expression in APs that was inhibited by SB431542. The MCF-7-induced expressions of α-SMA and FN in APs were also inhibited by SB431542. This indicated that TGF-β1 is necessary for BCCs-induced differentiation of APs into CAFs.

In summary, incubation of BCCs with APs increases TGF-β1 expression in APs. As observed in Figure 3, this is independent of the ciliation of APs. However, since TGF-β1 signaling is dependent upon the primary cilium in APs, when APs are deciliated, TGF-β1 signaling is not functional and cannot convert APs into CAFs.

### 3.4. Deciliation of APs Decrease Their Ability to Enhance BCCs Migration, But Not Proliferation

CAFs exert various effects on BCCs [2,3]. We investigated if the inhibition of APs differentiation into CAFs by cilium deletion modified some of these effects. First, we analyzed the ability of APs and CAFs to modulate cell proliferation. APs, grown on cover slips, were treated or not with HPI-4 and then incubated with BCCs. After four days, cover slips were removed and BCCs were counted (Figure 7A). As observed, incubation of MCF-7 or T47D with APs induced a small, but statistically significant, increase in BCCs proliferation. This increase was similar when BCCs were incubated with ciliated or unciliated APs. This indicated that deciliation of APs did not affect the ability of APs to stimulate BCCs proliferation.

CAFs are also known to increase BCCs migration [32,33]. We performed a wound healing assay to determine the effect of APs deciliation on BCCs migration. APs grown on cover slips were treated or not with HPI-4 and incubated with BCCs for four days to convert them into CAFs. Then, cover slips were removed. BCCs were re-plated in the presence of mitomycin C to decrease interference of cell proliferation on migration, and then scratched in depletion medium (an outline of the protocol is provided in Appendix A).

Pictures of the same field were taken immediately after the scratch and 24 h later (Figure 7B). CAFs increase BCCs migration. When APs cells were deciliated by HPI-4 before incubation with BCCs, the migration of BCCs was decreased. These results indicated that deciliation of APs inhibited their differentiation into CAFs and their ability to stimulate BCCs migration.

## 4. Discussion

The primary cilium is an important regulator of cell proliferation [34,35]. As a result, most cancer cells lose their primary cilium, which is a phenomenon that has been hypothesized to promote tumorigenesis and metastasis. In breast cancer, this loss is correlated with the degree of transformation [36]. Although cancer cells are usually not ciliated, most cells within the tumor microenvironment display a cilium. The importance of non-cancer cells of the microenvironment, and CAFs in particular, for tumor development is extensively studied [2,3,4]. CAFs exert several functions from extracellular matrix production to growth factors secretion. In breast cancer, CAFs are particularly abundant and produce the desmoplastic response: a dense collagenous stroma responsible for the clinical presentation of a tumor as a ‘lump.’ Because of the abundance of adipose tissue within the breast, APs are likely candidate to convert into CAFs in breast tumors. Obesity, which is associated with an enhanced number of adipocyte and of APs, is associated with an increased incidence and mortality of breast cancer [37]. Since the seminal paper of Karnoub et al., the conversion of mesenchymal stem cells into CAFs by BCCs in vitro and in vivo has been the subject of intensive studies. Here, we reveal that the primary cilium of human APs controls the conversion of APs into CAFs.

These results were obtained by using a simple co-incubation technique. One type of cell remains on cover slips while the other surrounds them. Thus, cells are constantly immersed in the secretome of both cell types. At the end of the experiment, cells can be separated and studied independently without trypsinization or cell sorting. Compared to conditioned medium, which only reveals the effect of a secretome of one cell type on another, this technique can be used to study potential exchanges between the two cell types. Thus, although rudimentary, this technique offers several advantages. One drawback is that direct cell-to-cell interactions are not revealed by this method.

Analysis of various cancers at the single cell level have revealed that, within the same tumor, CAFs are highly heterogenous cells that differ in gene expressions and functions [6,7,8,9]. Thus, some authors have used different classifications such as “myo CAFs,” “inflammatory CAFs,” “proliferating CAFs”, “vascular CAFs”. The cells studied in this manuscript present the characteristics of “myo CAFs” since the expression of four myofibroblastic genes. α-SMA, FN, COL1A, and TGF-β1 are increased after differentiation of APs into CAFs. Incubation of APs with BCCs increases TGF-β1 expression in APs in a cilium-independent fashion. This is in agreement with the observation that TGF-β1 production in human mesenchymal stem cells is mediated through the production of osteopontin by BCCs that binds to integrin on mesenchymal stem cells [38]. Since integrin, which is the receptor for osteopontin, is not located in the primary cilium, this organelle does not control TGF-β1 production in APs. Leptin expression has already been reported in CAFs from B-APs [39]. In our experiment, it was increased after incubation with BCCs and did not appear to be dependent upon the primary cilium. The expression of other CAF-related genes was not affected. PDGF-R α expression was not modified. However, PDGF-R α is already highly expressed in APs and is an established marker of APs in the adipose tissue [40]. FSP/S100A4 expression was not modified after co-incubation with BCCs. FSP is considered to be a marker for quiescent fibroblasts, rather than CAFs [5], and is not expressed in CAFs that express a-SMA [41]. FAP expression also was not modified. Although FAP is often highly expressed in the total CAFs population, it is only present in a fraction of CAFs [5,42]. In breast cancer specifically, Costa et al. have identified FAP in only one out of four populations of CAFs, while α-SMA was found in two [43]. 

Considering that CAFs are highly heterogeneous and that no definitive gene pattern expression for the various CAFs populations have been firmly established, it is not possible to determine if the “in vitro CAFs” obtained in our experiments represent truthful maturity in vivo CAFs. However, these cells present characteristics of CAFs since (A) they differentiate under the influence of BCCs, (B) the differentiation is dependent upon TGF-β1, an established in vivo inducer of CAFs, (C) they express several genes which have been found in “myo CAFs,” and D) the CAFs exert biological effects on BCCs.

The cilium has been previously involved in the differentiation of various cells such as neuronal cells, adipocytes, and in cells from the skin [16,44]. It also regulates branching morphogenesis during mammary gland development [45]. Here, we show that it is necessary for the conversion of APs into CAFs under the influence of BCCs. This is dependent upon a production by APs of TGF-β1 1 induced by BCCs. TGF-β1 1 is expressed at a high level in the plasma of breast cancer patients and plays a critical role in the differentiation of CAFs [2,3,4,30]. In APs, we, and others, have shown that TGF signaling is dependent upon the primary cilium [12]. It has been established that the TGF-β1 receptor is located in the primary cilium of various cells, and specifically in human mesenchymal stem cells [31]. Deciliation of APs makes them resistant to TGF-β1 and prevents CAFs differentiation.

Our results indicate that the cilium is a necessary element in APs conversion into “myo CAFs.” The question of the function of the cilium in the differentiation of other types of CAF is interesting. Besides APs, CAFs can be produced from other cell types such as fibroblasts, endothelial cells, or cancer cells through an epithelial to mesenchymal transition [2,3,4]. For instance, adipocytes co-incubated with BCCs differentiate into a special type of CAFs called “adipocyte-derived fibroblasts (ADFs)” and promote tumor progression [33]. ADFs do not express α-SMA and, thus, are not “myo CAFs” [2,3]. Since adipocytes do not possess a primary cilium, ADFs conversion from adipocytes is independent of this organelle.

To summarize (Figure 8), we demonstrate here that the primary cilium is necessary for the differentiation of APs into CAFs. BCCs increase production of TGF-β1 in APs. TGF-β1 is necessary for the differentiation of APs into “myo CAFs” expressing α-SMA, FN, Col1A1, and TGF-β1. These CAFs increase BCCs migration. In the absence of a cilium, APs still produce TGF-β1 in the presence of BCCs but, since TGF-β1 signaling is dependent upon the primary cilium, APs do not differentiate into CAFs and, thus, do not modify BCCs’ migration. This is the first observation of a function of the cilium of APs in their conversion into CAFs and its consequences on BCCs.

There is a documented link between CAFs number and function and the outcome of cancer [3]. Thus, these cells are potential targets for developing anti-tumor drugs. Several clinical trials that target CAFs activation are underway. Among the targets of these trials are Hh and TGF-β1 signaling. Hh signaling is dependent upon the primary cilium [16] and our results indicate that the primary cilium is necessary for the differentiation of APs into CAFs through inhibition of TGF-β1 signaling. Thus, by targeting the cilium in APs in breast tumor, two potent inducers of CAFs would be inhibited. However, one should keep in mind that the primary cilium has important biological function throughout the body and a global inhibition of the cilium could be deleterious.

## Figures and Tables

**Figure 1 cells-09-02251-f001:**
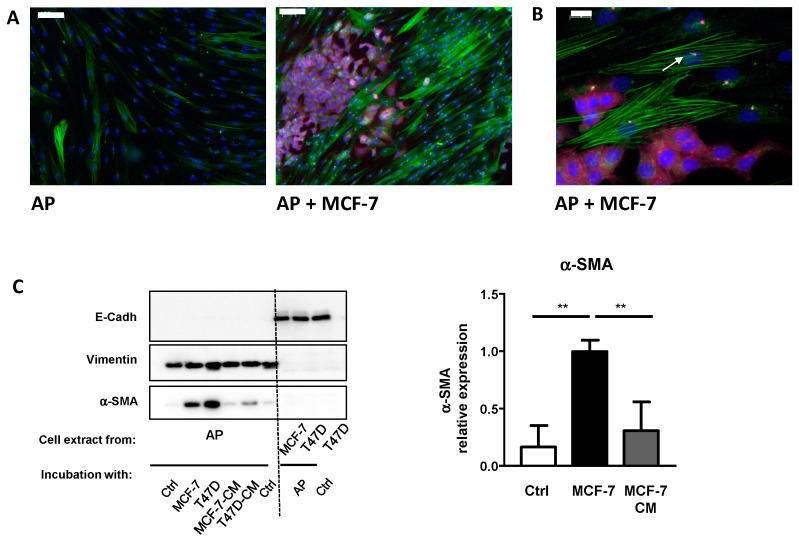
MCF-7 convert adipose progenitors (APs) into cancer-associated fibroblasts (CAFs). (**A**) Human APs and MCF-7 cells were co-cultured, fixed, and analyzed by immunocytochemistry using α-SMA (green), E-Cadherin (pink), ARL13B (green), and (**B**) Pericentrin (red). Nuclei were stained with Hoechst 33258 (blue). The white bar represents 100 µm in A and 20 µm in B. (**C**) Human APs on cover slips were co-incubated in the presence of control medium, MCF-7, or T47D cells or with conditioned medium (CM) from MCF-7 or T47D for four days. Proteins were extracted and analyzed by Western blot using E-Cadherin, Vimentin, and α-SMA antibodies. A quantification of α-SMA in three independent experiments is provided. ** *p* < 0.01.

**Figure 2 cells-09-02251-f002:**
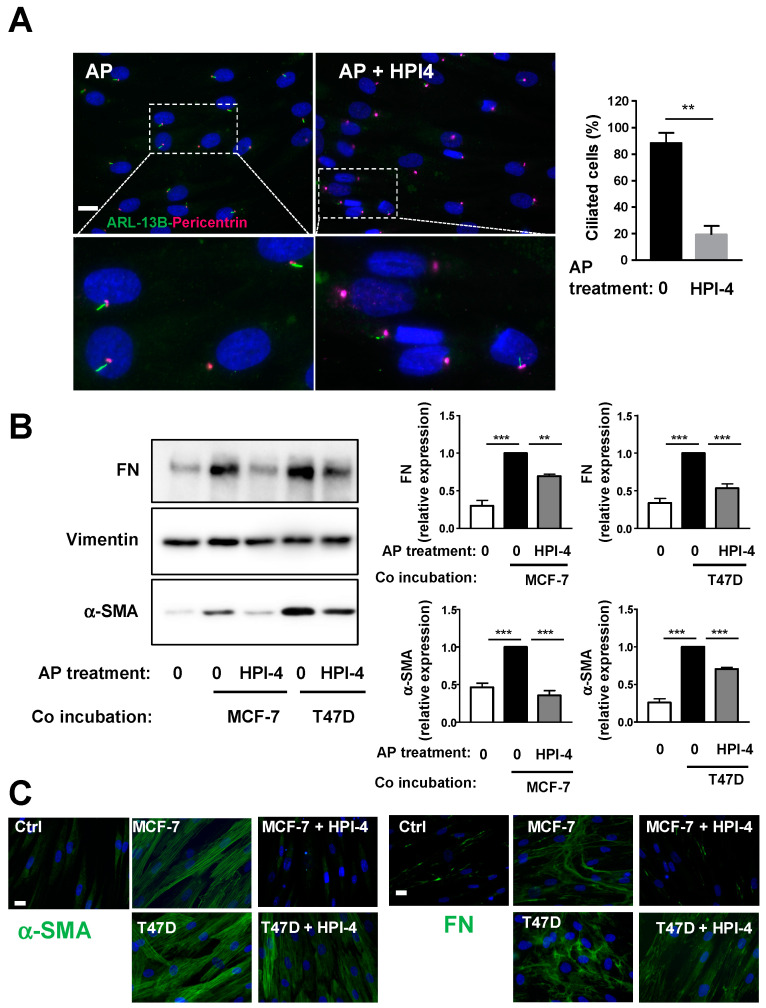
Pharmacological inhibition of the primary cilium inhibits conversion of APs into CAFs. Human APs on cover slips were treated or not with HPI-4 for 24 h (**A**) cells were fixed and ARL-13B (green) and Pericentrin (red) were revealed by immunocytochemistry, nuclei were stained with Hoechst 33258 (blue). A magnification is provided below each picture. The white bar represents 20 µm. Right: quantification of ciliated cells in different conditions. APs treated or not with HPI-4 were co-incubated in the presence of control medium (0) or with MCF-7 or T47D for four days. (**B**) Proteins were extracted, and Western blots were performed using the indicated antibodies. A quantification of four independent experiments is provided. ** *p* < 0.01 *** *p* < 0.001. (**C**) Cells were fixed and α-SMA or FN (green) were revealed by immunocytochemistry and nuclei were stained with Hoechst 33258 (blue). The white bar represents 20 µm.

**Figure 3 cells-09-02251-f003:**
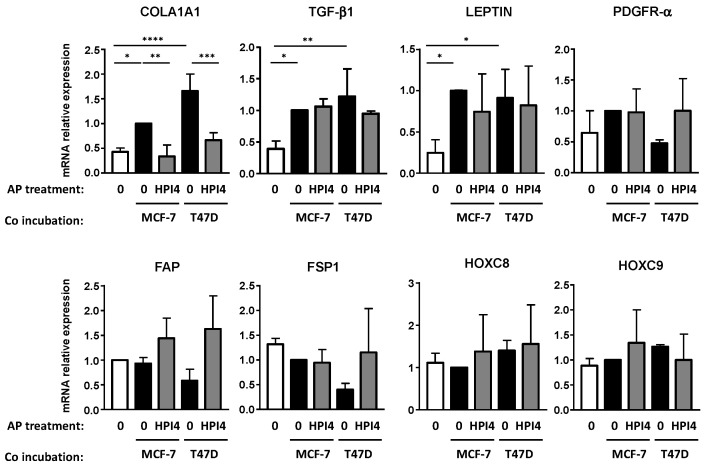
Pharmacological inhibition of the primary cilium decreases conversion of APs into “myo CAFs”. Human APs on cover slips were treated or not with HPI-4 for 24 h and co-incubated in the presence of control medium (0) alone or with MCF-7 or T47D for four days. RNA were extracted and quantitative RT-PCR were performed on the indicated genes. A quantification of three independent experiments is provided. * *p* <0.1, ** *p* < 0.01, *** *p* < 0.001, **** *p* < 0.0001.

**Figure 4 cells-09-02251-f004:**
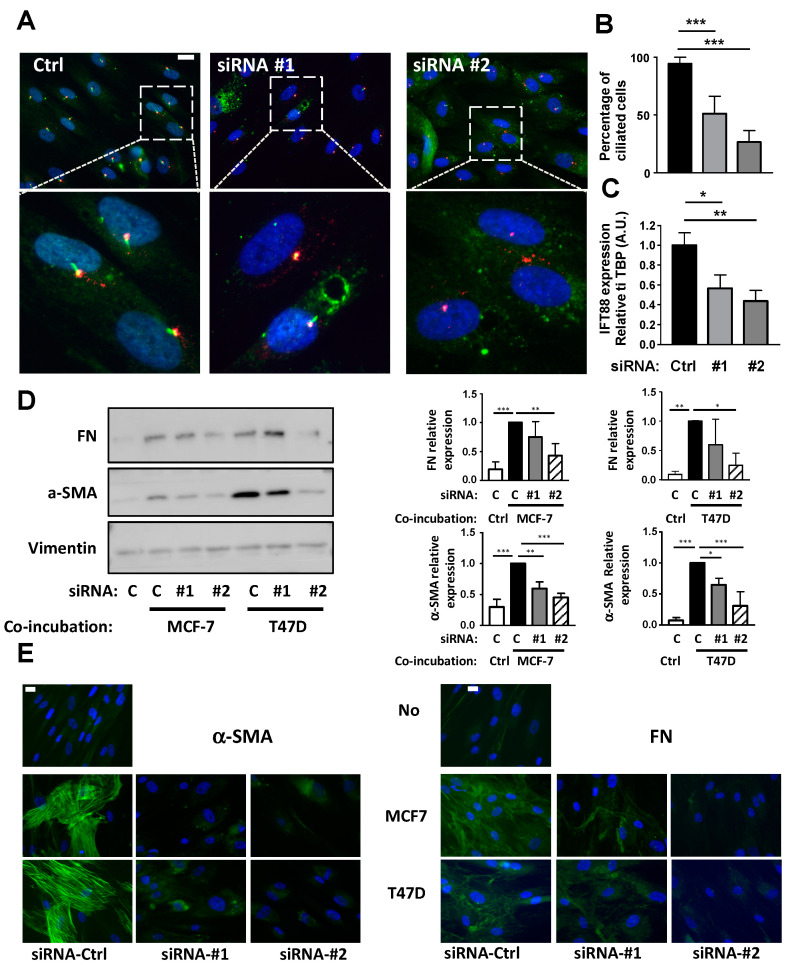
Inhibition of the primary cilium through siRNA against IFT88 prevents conversion of APs into CAFs. (**A**) AP were treated with a siRNA control (Ctrl) or against IFT88 (#1 and #2) as described in the methods. (**A**) AP were fixed and ARL-13B (green) and Pericentrin (red) were revealed by immunocytochemistry. Nuclei were stained with Hoechst 33258 (blue). A magnification is provided below each picture. The white bar represents 20 µm. (**B**) Percentages of ciliated cells in different conditions were calculated (**C**) mRNA for IFT88 were measured by quantitative RT-QPCR. APs were treated or not with a non-specific siRNA (ctrl) or siRNA #1 or #2 against IFT88 and co-incubated in the presence of control medium (c) or with MCF-7 or T47D for four days. (**D**) Proteins were extracted and Western blots performed using the indicated antibodies. A quantification of three independent experiments is provided. * *p* < 0.05, ** *p* < 0.01, *** *p* < 0.001. (**E**) Cells were fixed and α-SMA or FN were revealed by immunocytochemistry (green). Nuclei were stained with Hoechst 33258 (blue). The white bar represents 20 µm.

**Figure 5 cells-09-02251-f005:**
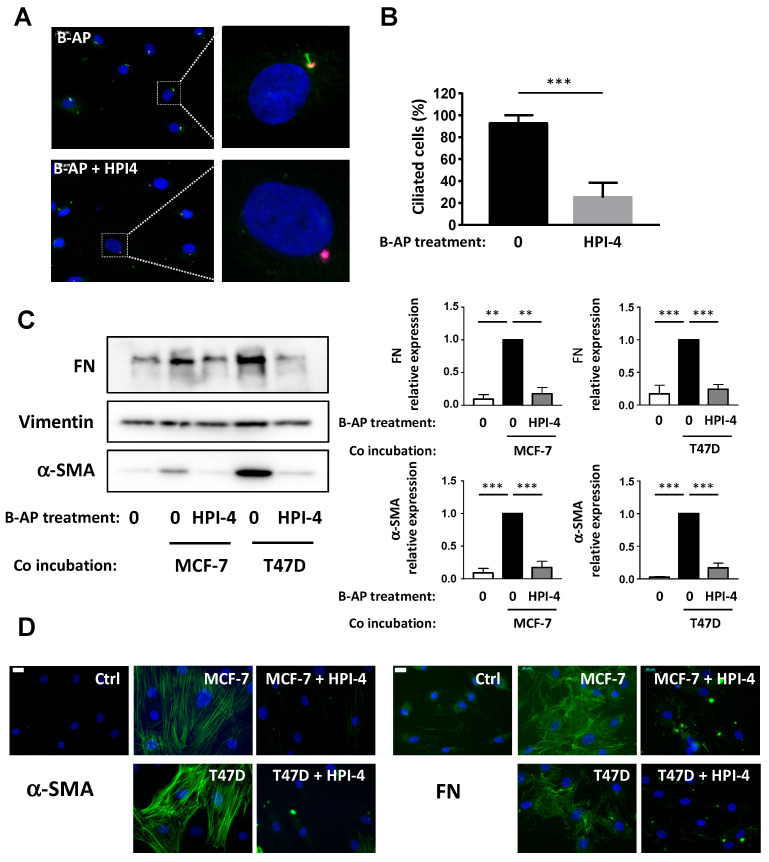
Pharmacological inhibition of the primary cilium inhibits conversion of primary culture of B-APs into CAFs. (**A**) primary culture from B-APs were treated for 24 h with HPI4. Cells were fixed and ARL-13B (green) and Pericentrin (red) were revealed by immunocytochemistry. Nuclei were stained with Hoechst 33258 (blue). The white bar represents 20 µm. (**B**) Percentages of ciliated cells in different conditions were calculated (**C**). B-APs were grown on cover slips and treated or not with HPI-4 and co-incubated in the presence of control medium (0) or with MCF-7 or T47D for four days. Proteins were extracted, and Western blots were performed using the indicated antibodies. Quantification of three independent experiments is provided ** *p* < 0.01, *** *p* < 0.001. (**D**) Cells were fixed and α-SMA or FN were revealed by immunocytochemistry (green). Nuclei were stained with Hoechst 33258 (blue). The white bar represents 20 µM.

**Figure 6 cells-09-02251-f006:**
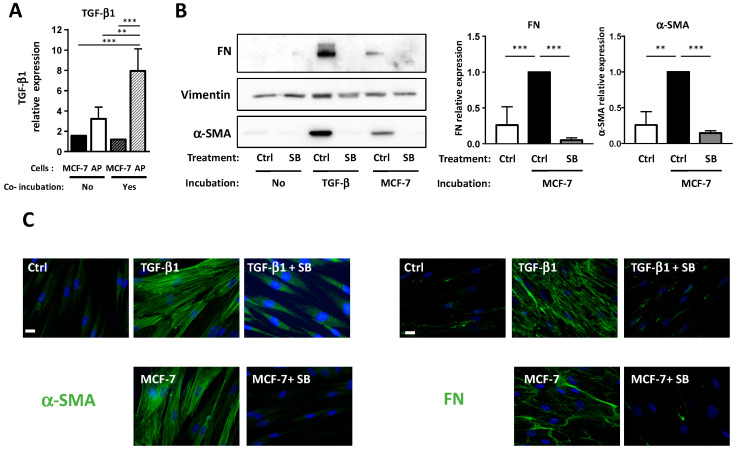
MCF-7 converts APs into CAFs through TGF-β1 production. (**A**) APs were grown on cover slips and incubated with a control medium or with MCF-7 for four days. TGF-β1 expression was assessed using quantitative RT-PCR in APs or BCCs alone or after co-incubation (**B**). APs were grown on cover slips and incubated for four days in the presence of control medium (ctrl) or with TGF-β1, or with MCF-7 in the presence or in the absence of SB431542. Western blots were performed using the indicated antibodies. Quantifications of three independent experiments are presented. ** *p* < 0.01, *** *p* < 0.001. (**C**) Cells were treated as described in B, fixed, and α-SMA or FN were revealed by immunocytochemistry (green). Nuclei were stained with Hoechst 33258 (blue). The white bar represents 20 µM.

**Figure 7 cells-09-02251-f007:**
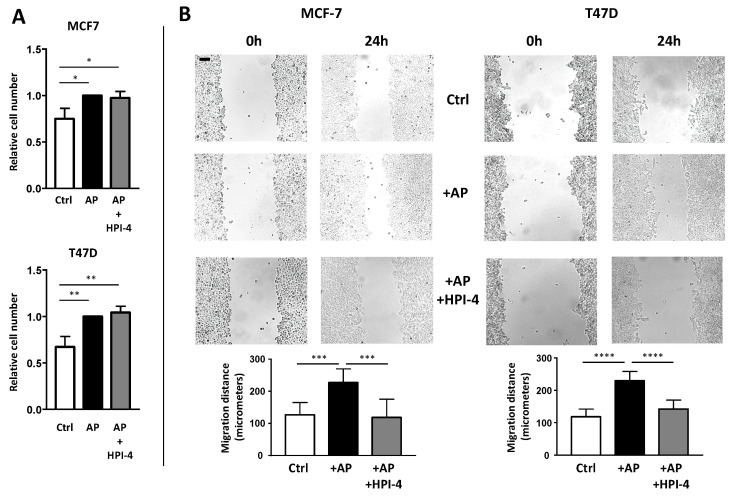
The primary cilium of APs is necessary for CAFs to promote migration of BCCs. MCF-7 and T47D were grown on dishes. Empty cover slips (ctrl) or coverslips containing APs treated or not with HPI-4 were added to the BCCs (**A**). After four days, cells were counted. A quantification of three independent experiments is provided. (**B**) After three days, BCCs were scratched as described in the Material and Methods section. Pictures at the exact same place were taken immediately and 24 h after the scratch. Eight different measures were performed for each condition. A quantification of a representative experiment (performed four times) is provided. The black bar represents 100 µM ** *p* < 0.01, *** *p* < 0.001, **** *p* < 0.0001.

**Figure 8 cells-09-02251-f008:**
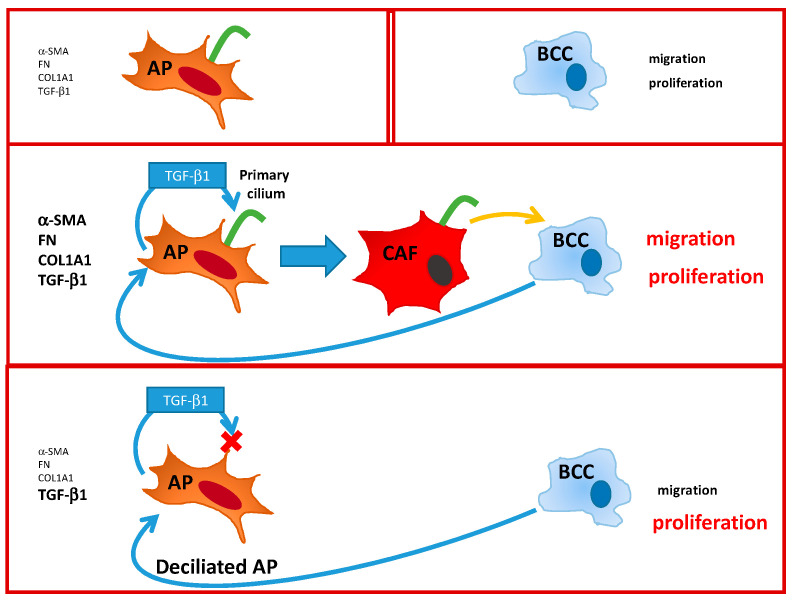
Summary of the interactions between Adipose Progenitor (AP), Breast cancer cell BCC, and Cancer Associated Fibroblast (CAF).

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
