# Peer review of "The Primary Cilium of Adipose Progenitors Is Necessary for Their Differentiation into Cancer-Associated Fibroblasts that Promote Migration of Breast Cancer Cells In Vitro"

_cells, 2020, doi:10.3390/cells9102251_

Round 1

Reviewer 1 Report

The Authors have added some new experiments in support of the first version of the manuscript, which appears now improved, overall.

As for this Reviewer’s request, they have measured TGF-beta1 expression in APs upon inhibition of cilium biogenesis, founding that it is not affected. This is apparently in disagreement with their previous statements of cilium involvement  in APs differentiation into CAFs (through the primary cilium ) via the TGF-beta1 signaling. However, they discuss these results suggesting that activation of the TGF-beta1 pathway through the primary cilium is dependent on an autocrine loop, in turn triggered by BCCs which induce TGFbeta1 expression via an osteopontin-integrin signaling. This is a quite complex scenario, which only becomes clear thanks to the summary proposed in Figure 8. This Reviewer suggests to introduce such model in paragraph 3.3, to help the understanding of the results.

Author Response

To meet the reviewer comment we have added a new sentence in paragraph 3.3 :

(lines 319-322)

“In summary, incubation of BCCs with APs increases TGF-b1 expression in APs. As observed in Figure 3 this is independent of APs ciliation. However, since TGF-b1 signaling is dependent upon the primary cilium, when APs are deciliated TGF-b1 signaling is not functional and cannot convert APs into CAFs. “

The effect of osteopontin on TGF-b1 production by APs are explained in more details in the discussion

Reviewer 2 Report

The paper has been enhanced with new experiments and new figures. The new data showing the differences in CAF markers expression is important to scientists in this field. I also appreciate the clarifications in the "material and methods", the extended discussion and the new cartoon.  I believe the paper is now suitable to be published in present form.

Author Response

We would like to thank the reviewer for his comments that have helped us to strengthen our work and for her/his kind evaluation of our revised manuscript.

This manuscript is a resubmission of an earlier submission. The following is a list of the peer review reports and author responses from that submission.

Round 1

Reviewer 1 Report

The primary cilium has received considerable attention during the last two decades regarding its involvement in cell signaling. More recently, it was also described in the oncological context. This manuscript addresses the interactions between cancer associated fibroblasts from adipose tissue and breast cancer cell lines. The latter can induce CAFs from adipose progenitors, a process that apparently depends on the primary cilium. Although the group has published in this context before, the rationale and conclusions remain somehow obscure.

My suggestion is to include a diagram explaining the interactions and molecules involved. TGFbeta signaling is known to affect breast cancer, the impact on primary cilia could be a secondary/side effect.

The degree of deciliation by siRNA should be measured in order to interpret the data.

What is the reason that BCC-conditioned medium and co-culture do not have similar effects (Figure 1)?

Furthermore, the manuscript lacks explanations and descriptions of the antibodies used. Their sources and specifications cannot be found in the methods section, and what exactly they bind to has not been explained sufficiently in the text. Considering the central importance of the primary cilia, the readers should be more convinced about the impact of inhibition on their presence. The presence and absence of primary cilia must be demonstrated in better pictures, ideally by TEM, but at least in high resolution confocal laser scanning microscopy with appropriate controls. 

Reviewer 2 Report

T&he article shows how primary cilium of adipose progenitors is necessary
 for their differentiation into cancer-associated fibroblasts that promotes migration of breast cancer cells in vitro. mThe study it is well performed, adequately controlled, with different approaches leading to the same result and statistical analysis of the results well done.

Overall, the set of experiments is quite limited but the research design appropriate, the methods adequately described , the results clearly presented and sufficiently conclusive.

One  missing experiment  is the demonstration that  inhibition of  cilium formation  prevents the increase of TGFbeta1expression  induced by co-culture .

Minor

Lane 205: (Sup Figure 2A) replace (Supplementary igure 2A)

Lane 247 : TGF-beta 1signalisation replace with "signalling"

Reviewer 3 Report

The manuscript from Peraldi et al. brings new data about the role of primary cilia in CAF formation from Adipose precursors. They tested their hypothesis with AP cell line and breast cancer-derived AP, showing an increase in fibronectin and alpha-sma when co-culture with BCC (or CM from BCC), and when cilia formation was inhibited they observed decrease in FN and alpha-sma. The group has a previous paper describing primary cilia as necessary for myofibroblasts differentiation on 2017.  The present paper is interesting, well-written and well-conducted, however here are my suggestions, doubts  and concerns about it:

1. I'd suggest that supplementary figures 1 and 2 should be placed onto the main manuscript. Figures are pretty, and in higher magnification (suitable for showing a small structure as cilia), I believe readers will benefit from having the proof that HPI-4 or siRNA is capable of disturbing cilia formation, close to the paper findings.

2. Why beta-actin is not used as a loading control on immunoblots?

3.  Have the authors tried to co-culture these cells on transwell systems?

4. Other papers have shown that alpha-smooth muscle actin also identifies different cell types derived from adipose progenitors (DOI: 10.1016/j.bone.2008.04.023 ; https://onlinelibrary.wiley.com/doi/pdf/10.1042/BC20110033). Other markers for CAFs should characterize these cells better (FAP/ PDGFRalpha/ S100A4, etc…). Fibronectin can be produced by any fibroblast (including myofibroblasts) and can not be used only with alpha-sma and vimentin to identify CAFs. Also, what happens with adipocyte markers (leptin, HOXC8, HOXC9, Tmem26 or Cd137)? Are they decreased when cells are secreting FN? Do they increase when cilia are inhibited? My question concerns if these cells (CAFs) are fully differentiated or not.

5. Fig. 6, Migration experiment is hard to understand.  A cartoon might help better in explaining this.

6. About CAF and BCC migration, it is not clear if HPI-4 was still in contact with cells and it's role in cell migration. Also, mitomycin C should be used to inhibit cell proliferation in wound healing experiments.

7. Fig. 5A- what about the TGFbeta levels on MCF-7 alone?

8. Although T47D appears to induce better the increase in FN and alpha-SMA (Fig. 2, 3, and 4), the authors did not use this cell line to test TGB-beta levels.  Why?

9. To affirm that "BCCs induce TGF-beta1 production in APs which binds to the primary cilium of APs leading to their differentiation into CAFs" more evidence should be shown, for example, the TGFbeta Receptor localization on primary cilia.